# The Vanishing and Renewal Landscape of Urban Villages Using High-Resolution Remote Sensing: The Case of Haidian District in Beijing

Hubin Wei [1,2] , Yue Cao [1,2] and Wei Qi [1,2,*]

1 Key Laboratory of Regional Sustainable Development Modeling, Institute of Geographic Sciences and Natural Resources Research, CAS, Beijing 100101, China
2 College of Resources and Environment, University of Chinese Academy of Sciences, Beijing 100049, China
* Correspondence: qiwei@igsnrr.ac.cn

**Abstract:** How to recognize the land use change in urban villages during dynamic transformation in Haidian District, Beijing, has become a hot topic with the promotion of urban renewal. The GF-1 high-resolution remote sensing images of 2013, 2015, and 2020 were used in this study to reflect the land use change in urban villages before and after urban renewal by using a hierarchical machine learning recognition method based on scene-based and random forest classification. The overall scale of urban village blocks in Haidian was 10.46 km$^2$, showing the distribution pattern along the traffic arteries in 2013. In 2015, it dropped to 10.11 km$^2$. The scale of urban village blocks in 2020 decreased to 1.02 km$^2$, 9.75% of that in 2013. Three kinds of urban village renewal logic are revealed by further taking Chuanying Village as an example: "urban village–blue–green space", "urban village–real estate", and "urban village–municipal facilities".

**Keywords:** urban village; urban renewal; land use change; scene-based; Haidian District; Beijing





## 1. Introduction

Informal settlements are a social issue that has attracted attention worldwide, which is highlighted in the United Nations' Sustainable Development Goals (SDGs) [1,2]. It is manifested through the slums in Latin America and self-help housing in the UK. In China, it turns out to be urban villages that arise because of the booming urban development [3]. In terms of function, these settlements have the attributes of rural areas while becoming migrant clusters because of their extraordinary location and relatively low cost of living and transportation. Thus, they can accommodate migrant labor from the countryside. In terms of architectural characteristics, urban villages show physical characteristics different from other urban buildings, such as a messy building style, high building density, and narrow roads. There are negative effects of these physical characteristics of urban villages arising from the rapid jump in urban expansion and the cost of land acquisition in urban development. First, there are many streets and alleys that are intricate and complex; so, entrance and exit routes are numerous and irregular, leading to fire and other safety problems. In addition, living and production facilities are relatively rudimentary, and domestic waste and sewage are rarely properly disposed of, leading to the spread of sanitation issues and epidemics in public places [4]. Furthermore, with the urbanization rate of China's population exceeding 60% in 2019, China has entered the middle and late stages of urbanization, and urban renewal and high-quality spatial development have become the new development goals. Therefore, urban villages have become the key blocks in urban renewal planning.

The emergence and development of urban villages can be traced back to China's economic transformation in the 1980s. From 1980 to 2017, the average annual growth rate of urbanization in China was 3%, which is 2.18% faster than the world average growth

rate [5]. It is the fastest and largest urbanization process in the history of the world. Amid the expansion of land use in cities, particularly in large cities, some rural areas are not incorporated into the unified construction of cities because of their high land acquisition costs or other complex historical and clan reasons. However, they remain under the villagers' self-governance system and collective ownership, becoming "gaping places" wrapped in the city. Given the unexpectedly rapid outward expansion of Chinese cities, these urban villages have gradually acquired excellent locations in just two decades [6]. However, their rental costs were lower than those of formal settlements because of their informal settlement properties. On the other hand, the contemporaneous relaxation of restrictions on the household registration system has increased the number of rural people coming to large cities seeking high income and opportunities. Given the low cost of living and the advantageous transportation location, the migrants choose urban villages for residence, and urban villages become migrant clusters [7]. In addition, the rapid expansion of cities cannot be achieved without a large number of cheap laborers. The migrant populations in the cities play an indispensable role in the construction and development of large-scale projects, whereas the informal settlements market in urban villages fills the gap in the demand for subsidized housing. Moreover, the informal settlements market becomes an important part of accommodating migrant populations and maintaining social stability [8]. The village and its members' needs for profitability, the migrant population's basic needs, including housing and living, and the needs of the urban economy and industrial development are the three levels of demands that drive and shape the village. Thus, it has become a hot spot for academic research.

In the study of urban villages in China, Guangzhou and Shenzhen in the Pearl River Delta play an undeniable role. Given the early acceleration of urban sprawl, suburbanization, and land expropriation, Guangzhou and Shenzhen were the first case areas where urban villages arose and received attention. The studies on the Pearl River Delta exceed the overall number of urban village studies by 50%. This topic has also received attention from developing countries in Southeast and South Asia, such as Vietnam and India [9,10]. In terms of research themes, informality is a crucial issue in urban villages [2,10]. Urban villages are considered the manifestation of informal settlements in China. Thus, the theories related to informal settlements are used to analyze the current situation and the development path of urban villages [11]. The informal settlements' market in urban villages based on location economy theory and land rent theory has become a hot topic for urban economics scholars, and both qualitative and quantitative studies on this topic have emerged [12,13]. This model of urban village development based on illegal buildings and the gap between the urban–rural dual system is also called informal development [14,15].

How can the drawbacks of informal development be reduced for the healthy development of urban villages? The urban renewal governance and transformation of urban villages have become the research focus in recent years. On the one hand, effective government–market coordination regeneration models have been summarized by considering and analyzing the existing regeneration cases, such as the neoliberal approach to coordinating with market forces in the regeneration process in Liede Village, Guangzhou [16]. On the other hand, the optimization of future related policies and systems, such as property rights policy, is also considered [17].

Spatial governance is a tool for social governance. In order to further the spatial governance of urban villages, spatial identification of urban villages has become a problem considered in remote sensing and GIS (Geographic Information System) disciplines. At the microscopic level, the footprint data and points of interest (POIs) are built using a density-based method to identify the height of buildings and building spacing in urban villages [18]. At the mesoscale level, Chen et al. proposed a hierarchical recognition framework that follows human cognition processes. It can also integrate remote and social sensing data to recognize fine-grained urban villages. The results showed good accuracy [19]. At a large scale, a partition-strategy-based framework was developed to improve recognition accuracy by repartitioning the city based on the characteristics of urban villages [20]. In

summary, the identification of urban villages has been studied at various scales. However, the static architectural features, spatial features, and proximity features of urban villages were detected well. Moreover, the dynamic spatial and temporal patterns of urban villages were not studied for a certain time series span.

Land use and cover change (LUCC) is beneficial to reflect the dynamic patterns of change in urban villages. LUCC has been an enduring research topic in environmental sciences, ecology, and geography since the 1990s [21]. In terms of the evolutionary mechanisms, the two main themes are the historical evolutionary characterization and the future-oriented spatiotemporal modeling. The former uses a time-series synthesis of multiple indicators to analyze the volume, rate, magnitude, and speed of land use change, whereas the latter uses various models, such as CLUE-S and CA, to predict future trends in land use scale and spatial pattern development [22]. The fundamental cause of LUCC evolution is the driving mechanism of LUCC, which can be mainly divided into natural and socioeconomic drivers [23]. In view of Tobler's First Law of Geography, the larger the scale of the spatial territory is, the more obvious the differences in the driving mechanisms of LUCC and the main components of its effects are [24]. Thus, large-scale LUCC is a constant hot spot for research. By contrast, small-scale LUCC is rarely studied. Given the availability of data and the low cost of research, land use types with remarkable natural and social characteristics are always the object of study and discriminative features of LUCC. However, semantically complex urban spaces, such as urban villages, are seldom studied owing to their rapid renewal and many changes. What is the flow of land in the urban village area after renewal? How does this change? These questions need considerable attention in the small-scale and semantically complex space of urban villages.

The identification of urban villages is an enormous challenge in complex urban environments [25]. The traditional pixel-based extraction methods of remote sensing information are not applicable to urban village extraction, since individual urban village pixels do not differ much from other urban buildings [26]. Therefore, it is necessary to exploit the semantic information surrounding urban villages in order to extract it accurately. Extensive research on high-resolution remote sensing information extraction tends to obtain semantic objects with similar spectrum or texture information by semantic segmentation of images, which leads to the classification of these objects. Although this object-based approach is effective for distinguishing vegetation, buildings, and other landscape forms, urban villages and urban building objects with similar characteristics cannot be further distinguished [19]. Here, the scene-based method shows strong superiority in urban village identification. The base classification unit of the scene-based approach is not a single object but a scene that aggregates information of multiple semantic objects [27]. Because of the particular historical background, the urban village scenes are vastly different from the traditional urban building scenes in terms of general patterns and building density, which also shows certain regional variations [28]. Consequently, it is effective to identify urban villages through the following hierarchical steps: First, urban building objects are extracted through an object-based approach. Then, the semantic information (spectrum, texture, geometry, etc.) of all building objects in the urban area scene is statistically calculated. Finally, machine learning methods are applied for identification.

Machine learning methods show the advantages of high accuracy and speed compared with traditional manual calibration. Machine learning methods in existing studies have many applications [29]. They are concentrated in areas such as agricultural remote sensing, and the research mainly focuses on urban climate topics, such as urban heat effects [30]. However, it is slightly directed at urban residential areas in the case of cities. Furthermore, random forest (RF) can effectively generalize high-dimensional data without feature selection [31]. Thus, it performs well on datasets. Rodriguez-Galiano et al. proved the robust performance of the RF classifier for the land cover classification of a complex area [32]. Given the superiorities of the RF method mentioned above, the present study uses scene-based RF classification for urban village identification.

In summary, this study uses the three phases of high-resolution remote sensing images of 2013, 2015, and 2020 and a hierarchical machine learning identification method based on the RF classification of objects to reflect the land use changes in urban villages before and after urban renewal. Thus, it can provide a reference for urban renewal decisions in Beijing.

## 2. Study Area and Data

### 2.1. Study Area

Haidian District is located northwest of the main urban area of Beijing, the capital of China. It comprises 29 administrative districts with a total area of 431 km², as shown in Figure 1. It also had a population of 3.133 million in the seventh census in 2020. In 2020, the gross regional product of Haidian District was CNY 850.46 billion, an increase of 5.9% year-on-year. The total economic volume and growth contribution are in first place in the city, due to the high-tech industries gathered in the area and the largest university community in the country. Given the rapid economic growth and the central position of urban renewal in the development pattern of the Beijing–Tianjin–Hebei innovative urban agglomeration, urban renewal with the goal of integrating resources and optimizing spatial functional patterns has brought about changes in the land use pattern of urban villages in Haidian District. These changes are difficult to ignore.

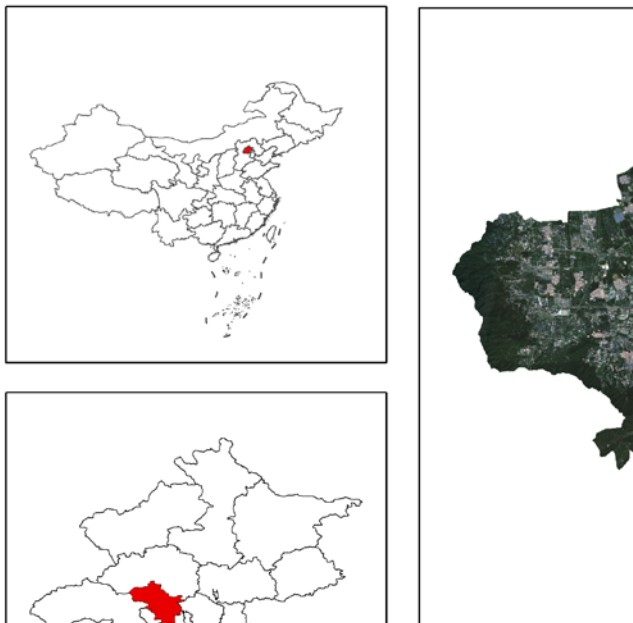

**Figure 1.** Location of Haidian District and street view.

### 2.2. Study Object

Urban villages are characterized by narrow roads, numerous entrances and exits, high building density, and limited area for public facilities. A notable distinction is observed between Beijing's urban villages and those located in southern Chinese cities such as Guangzhou and Shenzhen, which exhibit unique regional features that are reflected in their remote sensing images and architectural attributes. Furthermore, the urban villages in Haidian District can be classified into three distinct types in terms of architectural characteristics and their remote sensing images, as illustrated in Figure 2.

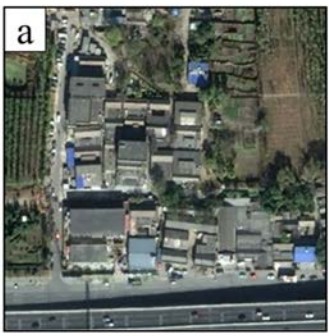
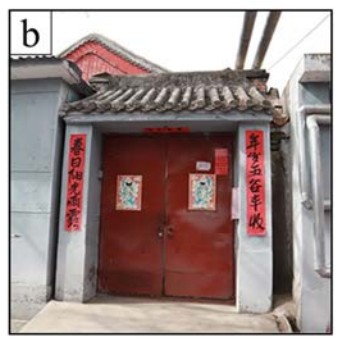

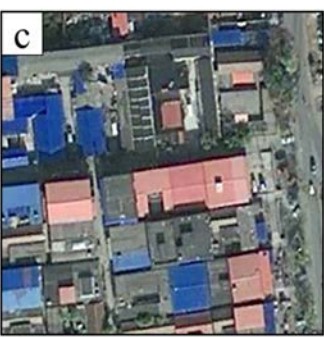
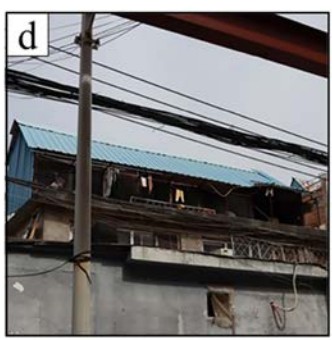

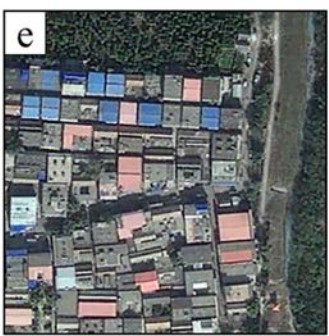
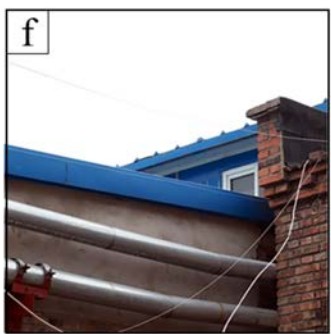

**Figure 2.** Three types of urban villages in Haidian District, Beijing: (**a**) remote sensing images of Type 1 urban villages, (**b**) architectural characteristics of Type 1 urban villages, (**c**) remote sensing images of Type 2 urban villages, (**d**) architectural characteristics of Type 2 urban villages, (**e**) remote sensing images of Type 3 urban villages and (**f**) architectural characteristics of Type 3 urban villages.

Type 1: The traditional Beijing residential courtyards as the main part. This type of urban village is usually located in the relative core of the city and is represented on remote sensing images as a cluster of black roofs and buildings with courtyards in a square layout.

Type 2: Shantytown as the main part. This type of urban village is located at the edge or periphery of the traditional urban area, and the remote sensing images of urban villages mostly show blue and red alloy roofs and chaotic road systems. It is worth mentioning that this type of urban village is gradually decreasing as the governance of urban villages in Beijing is enhanced, which is rarely observed in the remote sensing images of 2020.

Type 3: A combination of traditional courtyards and shantytowns. In this type of urban village, although traditional courtyards were built before shanty houses chronologically, they don't show a core-edge relationship spatially. Due to the dilapidated traditional houses or the encroachment of farmland, the shanty houses and courtyards may also show a hybrid layout. Therefore, the remote sensing image characteristic is also a combination of the above 2 types, and the overall amount of urban villages in this type is also the highest. Additionally, the urban villages in Haidian District have been equipped with heating pipes

since 2017, in accordance with Beijing's clean energy heating policy. The use of natural gas, a clean energy source, has replaced coal, thereby contributing to environmental protection. However, this has greatly impacted the streetscape of the urban villages, as illustrated in Figure 3. Unlike in ordinary urban buildings where heating pipes are placed inside, the pipes in the urban villages are erected on the exterior of the buildings. This not only affects the passability of streets in urban villages but also highlights a significant difference in the physical characteristics of urban villages in northern China, represented by Beijing, and those in southern cities such as Guangzhou and Shenzhen, in terms of their architectural features, as showed in Figure 3c,d [33,34].

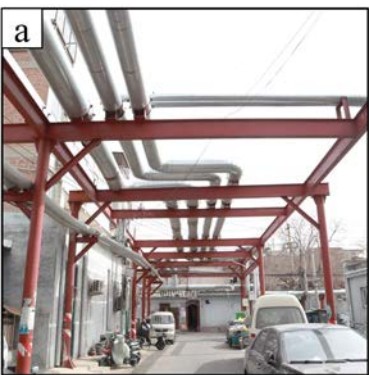
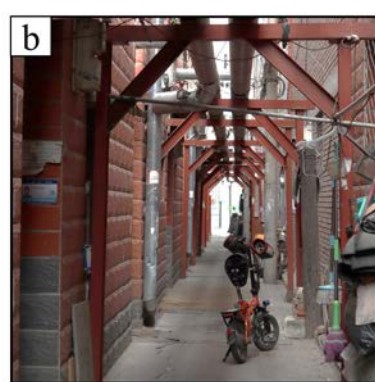

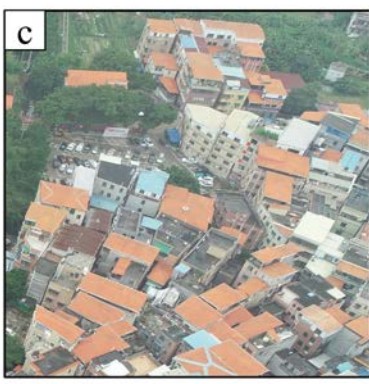
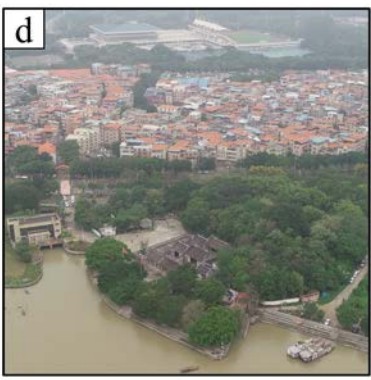

**Figure 3.** Comparison between the streets and alleys of urban villages in Beijing and those of urban villages in Guangzhou: (**a**) heating pipes in urban villages, (**b**) alleys in urban villages in Beijing, (**c**) alleys in urban villages in Guangzhou and (**d**) an aerial image of an urban village in Guangzhou.

### 2.3. Datasets

#### 2.3.1. GF-1 Satellite's High-Resolution Remote Sensing Images

GF-1 satellite is well-known for its consistently good signal, clear imaging, and widespread adoption by various industries, such as meteorological observation, forestry resource monitoring, epidemic assessment, traffic refinement management, etc. Launched in April 2013, GF-1 has a revisit cycle of 4 days for one satellite. Its hyper cameras PMS1 and PMS2 images include 8 m multispectral and 2 m panchromatic images. The 8 m multispectral includes four bands: blue, green, red, and near-infrared bands [35]. In this study, the high-spatial-resolution GF-1 satellite images acquired on 10 August 2013, 12 October 2015, and 9 August 2020, were applied. The scene in the images covered the entire Haidian District.

#### 2.3.2. Open Street Map (OSM)

Urban villages are spatially separated from urban areas because of the high number of roads around urban villages. Thus, urban village areas can be considered single inner-city blocks. This natural semantic information can be represented using OSM. This open-source

data map provider offers historical and current road network data [36]. It is characterized by a wide range of data sources and the high accuracy it brings with data labels corresponding to data attributes. Weiss, D. et al. pointed out that new data sources provided by OSM capture transportation networks with unprecedented detail and precision nowadays [37].

However, 2013 predates the earliest time of the historical data available in OSM. Thus, the 2013 scenario delineation uses the 2015 road data for consistency reasons, thereby increasing the accuracy of the 2013 block delineation. Therefore, the 2015 and 2020 OSM data are used for the scene delineation of Haidian District in this study.

## 3. Method

Urban villages are a unique phenomenon in China's urbanization process. They are located in the interior of urban areas. However, they often present a different scene from urban architecture. In contrast to the external characteristics of urban villages, which are usually adjacent to highways and surrounded by relatively neat and orderly buildings, the interior of urban villages is characterized by a lack of appropriate planning. In particular, the buildings have different styles and small footprints. Moreover, the areas are highly dense and exhibit close spacing, random distribution, and poor internal environments as shown in Figure 4 [38].

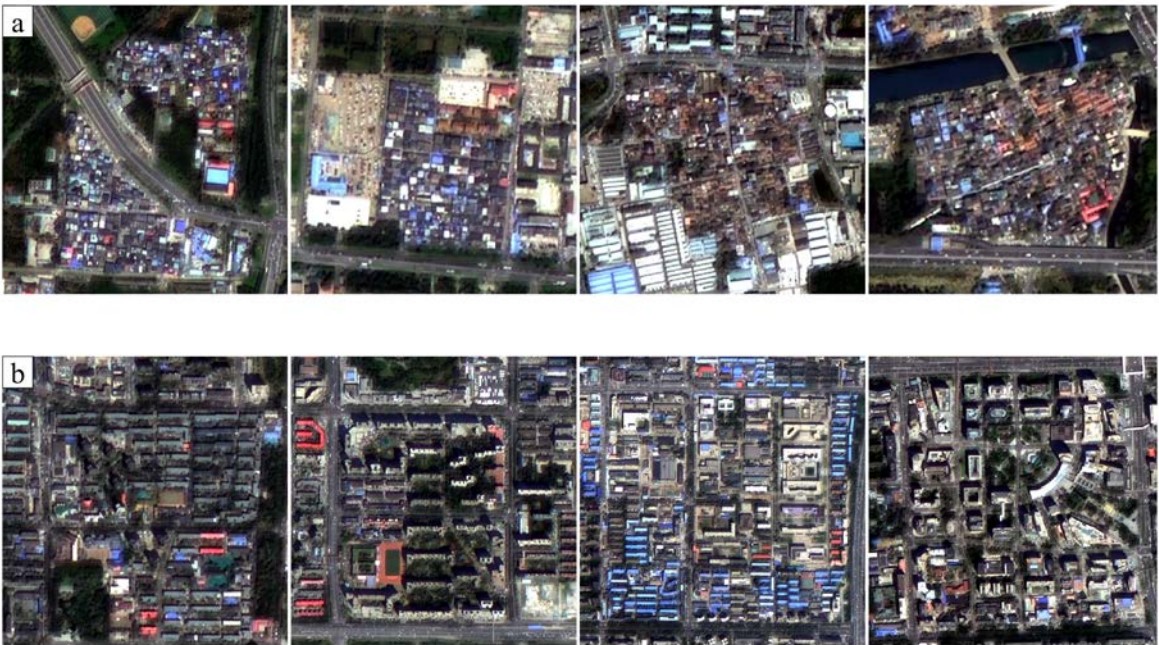

**Figure 4.** Sample examples of urban village blocks and non-urban-village blocks: (**a**) urban village blocks and (**b**) non-urban-village blocks.

Although the buildings in urban villages are very different from other buildings in the city, such differences are not always specific. The individual urban village building objects in remote sensing images do not differ much from factory houses and street facilities, which are not easily distinguished by traditional remote sensing methods. Moreover, extracting urban villages using high-resolution remote sensing images is challenging. Therefore, some scholars have proposed classifying urban village areas as a whole, i.e., the scene-based method, to identify urban villages [39]. This method makes full use of the overall characteristics of the urban village area, such as the high building density, small scale, and different styles. Thus, the confusion between individual urban village buildings, factories, and other buildings can be avoided. The urban villages can also be extracted with high accuracy. This study identifies the urban village areas in Haidian District, Beijing, in 2013, 2015, and 2020 based on the scene using the high-resolution remote sensing images from the DF-1 satellite, as shown in Figure 5. In the meantime, object-based classification methods

were used to extract three different land use types - urban buildings, roads, and blue–green space from remote sensing images.

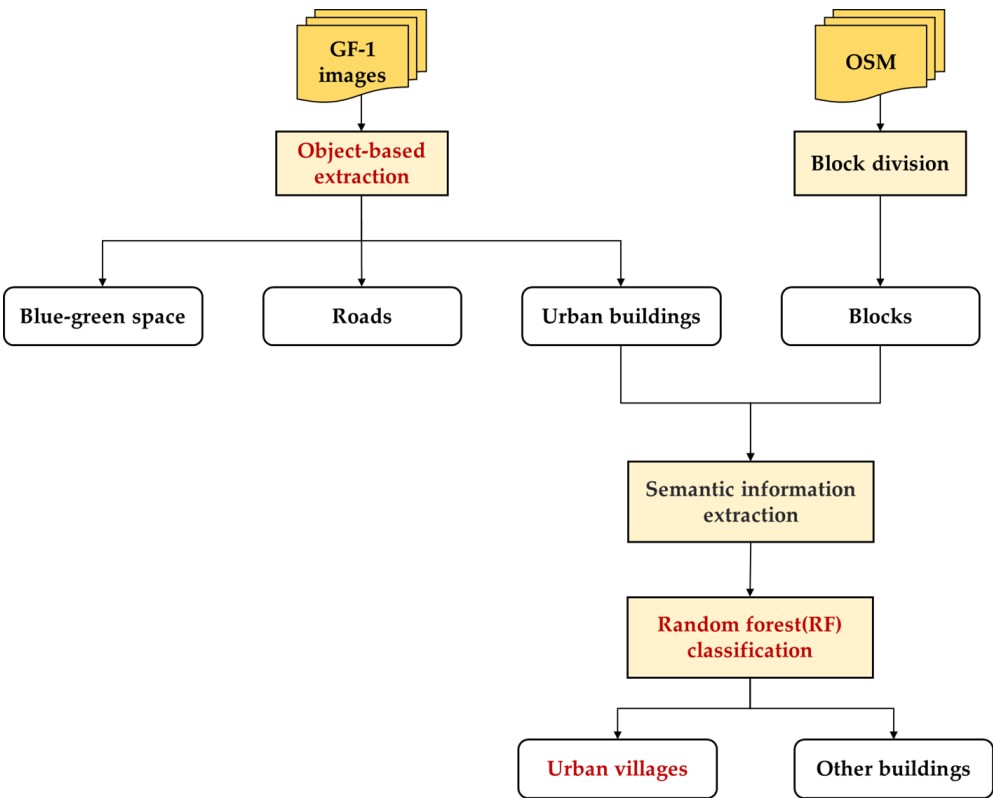

**Figure 5.** Workflow of urban village identification.

### 3.1. Scene Segmentation

Given the spatial fragmentation of urban village blocks from other areas of the city and the numerous surrounding roads, OSM was used to characterize the natural semantic information of urban villages. In this study, the OSM data from 2015 and 2020 were used to classify the scenes in Haidian District. The results are the minimum identification unit blocks of urban villages. Moreover, the OSM may be missing or wrong when the scenes are divided. Thus, we manually adjusted the OSM data of Haidian District in 2015 and 2020 to make the scenes reasonable by making up dead-end roads.

### 3.2. Building Information Extraction

The preprocessing of the GF-1 high-resolution remote sensing images was performed on ENVI. This preprocessing mainly included radiometric calibration, atmospheric correction, geometric correction, image fusion, mosaic, and cropping. The example-based feature extraction function in ENVI was used to extract the building object information from the processed images, which include each object. The information includes texture, spectral, and geometric information.

On the basis of the above remote sensing features, the area was classified into three types of land use, namely, urban buildings, roads, and blue–green space. Urban buildings refer to buildings in the city, including residential areas, public infrastructure, storage land, and urban villages. Roads are linear or radial roads used for transportation. The blue–green space includes woodland, grassland, and water, comprising a complex space, as shown in Figure 6.

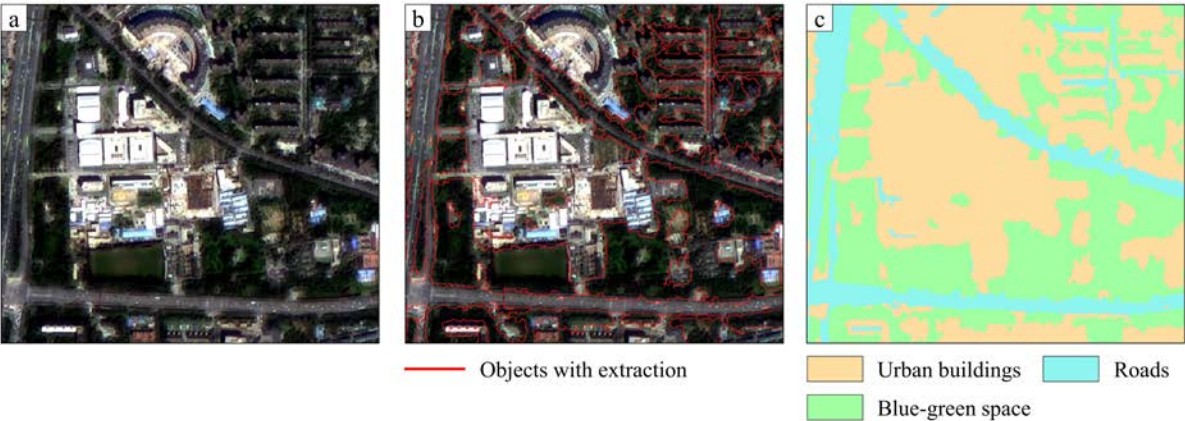

Figure 6. Building information extraction: (**a**) local high-resolution images, (**b**) image objects, (**c**) results of the building information extraction.

### 3.3. Scene-Based Urban Village Recognition

For each block scene, we selectively counted the information of building objects inside the scene, including the number of objects in the scene, the average area of the objects, the total area of the objects, the average of the object texture kernel, and other available object texture, spectral, and geometric information. This statistical information comprises the features of the neighborhood scene. It was used as the input to the RF method to identify the urban village scene.

The RF method is a classical method of machine learning. It is widely used in feature extraction. Moreover, it is a classifier containing multiple decision trees, and its output class is determined by the plural of the classes output from all decision trees. For each tree, it uses a training set sampled back from the total training set, i.e., some samples in the total training set may appear multiple times in the training set of a tree or they may never appear in the training set of a tree. In the training of the nodes of each tree, the features used were randomly drawn from all the features in a particular proportion without delay. The algorithm continued as follows:

- Randomly sample the training set $m$ times to obtain a sample set $D$ with sample size $m$.
- Randomly select $k$ attribute features among all attribute features and select the best segmented attribute features as nodes to construct the decision tree.
- Repeat the above two steps $T$ times, i.e., $T$ decision trees are built.
- These $T$ decision trees form an RF and decide which category the data finally belongs to by voting.

### 3.4. Accuracy Verification

In this study, the confusion matrix was constructed to verify the accuracy of urban village mapping. Each year, 100 randomly generated validation sample points were identified as urban villages by combining visual discrimination on Google HD images and field research. Then, the confusion matrix was constructed by combining the identification results to calculate the overall accuracy (OA) and consistency evaluation of the classification. The confusion matrix was calculated, as shown in Table 1.

**Table 1.** Confusion Matrix.

| Confusion Matrix | | True Value | |
|---|---|---|---|
| | | **Positive** | **Negative** |
| Predicted Value | Positive | TP | FP (Type II) |
| | Negative | FN (Type I) | TN |

TP includes the samples of urban villages with the correct classification (predicted as 1 and actually 1). FN comprises the samples of urban villages with misclassification (predicted as 0 but actually 1). FP includes non-urban-village samples with misclassification (predicted as 1 but actually 0). TN comprises the samples of correctly classified non-urban-village areas (predicted as 0 and actually 0).

The formulas for calculating the OA of the classification and the *Kappa* coefficient are as follows:

$$\text{OA} = \frac{1}{N}\sum_{i=1}^{r} x_{ii} \tag{1}$$

$$Kappa = \frac{N\sum_{i=1}^{r} x_{ii} - \sum_{i=1}^{r}(x_{i+} \times x_{+i})}{N^2 - \sum_{i=1}^{r}(x_{i+} \times x_{+i})} \tag{2}$$

In the formulas, $N$ is the total number of samples, and $r$ represents the number of classifications. In this paper, $r = 2$, $x_{ii}$ represents the number of samples correctly classified in class $i$, and $x_{i+}$ and $x_{+i}$ represent the number of real reference pixels and the total number of classified pixels in class $i$, respectively.

## 4. Spatiotemporal Distribution

In terms of the overall identification results and precision, the number of urban villages extracted in 2013, 2015, and 2020 was 28, 27, and 9, respectively, and the overall precision was 89%, 90%, and 95%, respectively. The *Kappa* coefficients were all above 0.8. The specific spatial and temporal distribution patterns are shown in Figure 7.

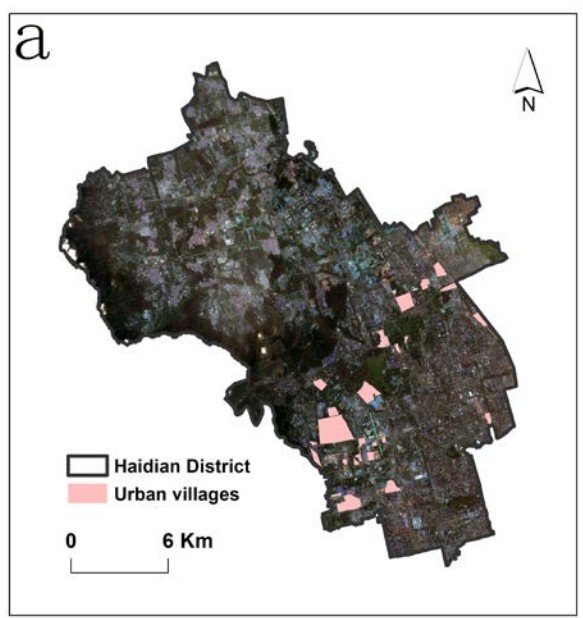
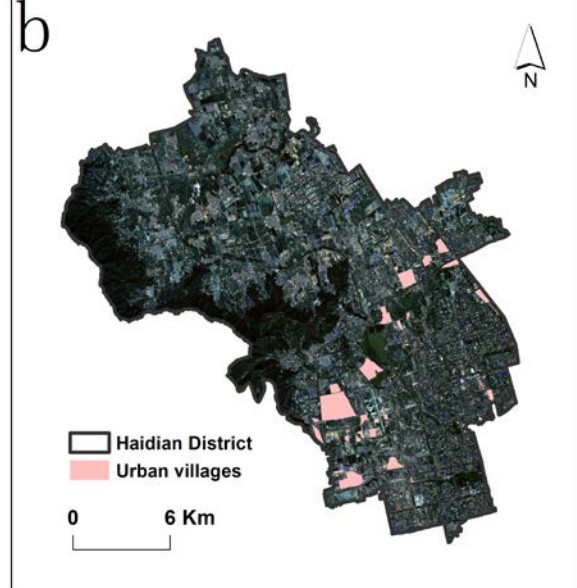

**Figure 7.** *Cont.*

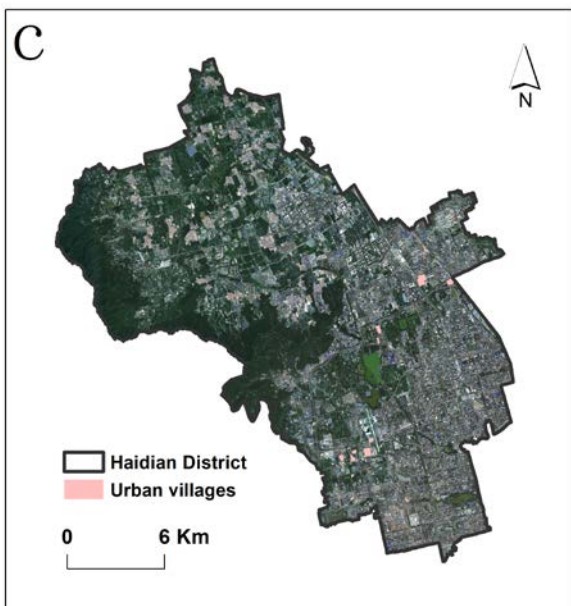

**Figure 7.** Overall results of the urban village recognition in Haidian District: (**a**) 2013, (**b**) 2015, and (**c**) 2020.

### 4.1. Spatial Distribution

In 2013, the urban village blocks covered a total area of 10.46 km$^2$, showing a clear spatial distribution pattern along traffic arteries. The urban village in the Haidian Subdistrict in the south of Haidian District had the largest area of 3.22 km$^2$. It was located east of the West Fifth Ring Road, one of the most important expressways in Beijing. The "four towns behind the mountains" (Wenquan Town, Sujiatuo Town, Xibeiwang Town, and Shangzhuang Town in the northwestern part of Haidian District), which were historically traditional agricultural and ecological areas, did not have urban villages. The urban villages in this period were mainly distributed around the West Fifth Ring Road and the North Fifth Ring Road, with a block area of 8.22 km$^2$, reaching 78.59% of the total urban village area. The remaining 21.41% of the urban village blocks were distributed around other major traffic arteries in Beijing, such as the Jingzang Expressway, South College Road, and Yongdinghe Road.

In 2015, the area of the urban village block was 10.11 km$^2$. It decreased at an annual average of 0.18 km$^2$ in 2 years. The renewal of the Xiangquanqiao block, located at the junction of the West Fifth Ring Road and the North Fifth Ring Road in the central Haidian District, was the dominant change in this period. Moreover, the area around Beijing North Railway Station in the southeastern Haidian District was a typical infrastructure-type urban village, indicating that infrastructure development agencies did not have a good connection with the surrounding land plans after acquiring infrastructure land for development. This scenario resulted in the emergence of infrastructure land–fringe urban villages. Therefore, although this block was one of the closest blocks in Haidian District in terms of location to the "Core Area of Capital" of Beijing, the urban village regeneration of Beijing North Railway Station lagged compared with that in the neighboring blocks.

In 2020, only 1.02 km$^2$ of urban village blocks remained, and the scale of the urban village renewal in 5 years was 9.09 km$^2$. Compared with the urban village blocks in 2013, 90.25% of the urban village blocks in 2020 underwent urban renewal. In 2020, only a few urban village blocks remained in clusters in the south of Qinghe Subdistrict, the middle of Qinglongqiao Subdistrict, and the south–central part of the Sijiqing Subdistrict, which were mainly located near large transportation hubs and administrative area junction lines. In terms of renewal flow, most of the transformed urban villages were replaced by green parks and real estate, among which blue–green space was the main flow of urban village

renewal in this period. Moreover, the urban villages in Chuanying, Xiaoyuehe, and Shucun blocks were transformed into urban corner parks, such as Shuying Park, Tayuan Urban Forest Park, and Shucun Park, respectively, probably as a result of the high demand for a beautiful environment and low-pollution living from educated residents in the district. By contrast, urban villages, such as Liaogong Village and Tian Village, located in areas with high land prices and good accessibility, were developed into real estate. This change was also related to the industrial development and rising employment scale of Haidian District.

*4.2. Quantitative Analysis*

In terms of the change trend of urban villages in Haidian District from 2013 to 2020, to be noticed first is the change in the quantity of urban villages and their block sizes, as illustrated in Figure 8. During 2013–2020, a total of 19 urban villages and 9.44 km$^2$ of urban village blocks were renewed into other landscapes due to Beijing's gradual urban renewal planning. On average, 2.71 urban villages vanish and 1.35 km$^2$ of urban village blocks were upgraded each year. This size is equivalent to 3214.29 basketball courts. Thus, the area of the urban village blocks in Haidian District is being converted to other landscape types in the rate of 459.18 basketball courts per year. The extensive conversion of urban village blocks in Haidian District has undoubtedly brought about a notable change on the district's function and for living and ecology [40].

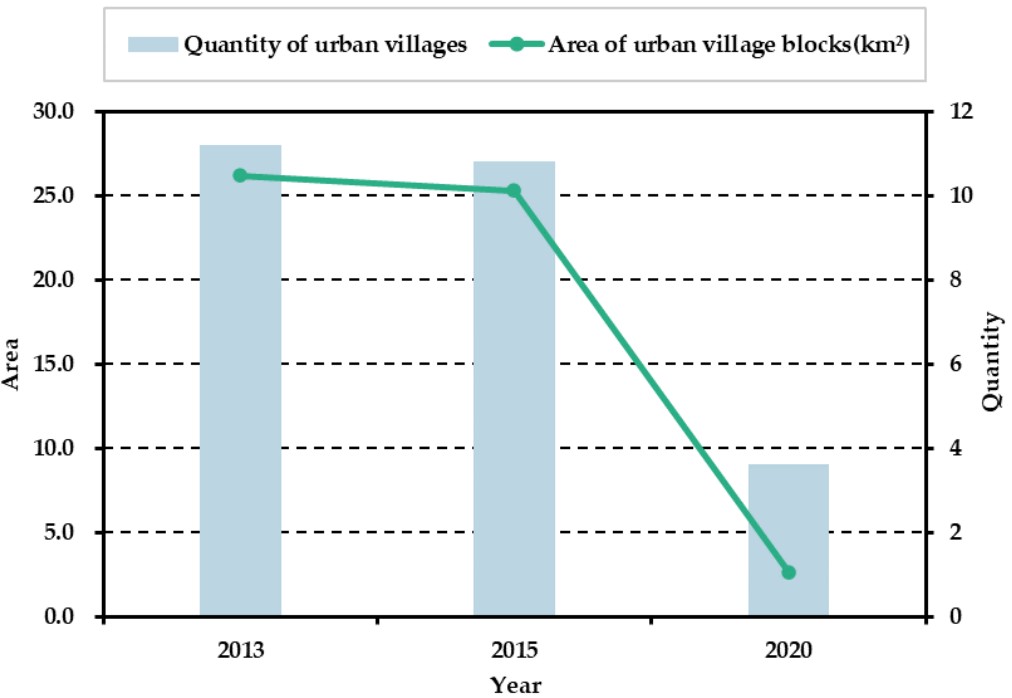

**Figure 8.** Trends in the quantity of urban villages and their block areas in Haidian District (2013–2020).

While the very rapid decline is shown in the bar chart between 2015 and 2020 compared to the slow change from 2013–2015, it is important to take into account that this rapid decline shows the overall change over a 5-year period, rather than the superficially average interval between the three periods shown in the chart. On average, 3.6 urban villages and 1.82 km$^2$ of urban village blocks vanish between 2015 and 2020 each year. This rate of change is only about 30% higher than the average rate between 2013 and 2015 (32.63% and 34.81% for the quantity of urban villages and the size of urban village blocks, respectively).

In addition, the progressively tighter number and size of urban villages demonstrates the guiding force of planning policies. Unlike Shenzhen's urban villages from 1999–2009 which saw a significant increase in land and built-up area size over the decade, urban village renewal in Haidian District, one of the regions where the core of China's capital is

located, has not only restrained the informal economy of urban villages, but also brought the interests of urban villages more under supervision [41].

From 2013 to 2020, the area of urban villages in Haidian District showed a continuous decline not only in terms of numbers but also in terms of regions. Unlike Shenzhen, no new urban villages in Haidian District were created in the 7 years. Moreover, no other urban spaces were transformed into urban villages in 7 years. This scenario is attributed to Beijing's relatively advanced level of urbanization and the government's emphasis on spatial planning since its early years. Its land use changes show a typical urban renewal logic of "urban village–blue–green space", "urban village–real estate", and "urban village–municipal facilities", which serve as the overall spatial planning of Beijing very well.

## 5. Case Study of Chuanying Village

In the land use changes in urban villages, three kinds of urban regeneration logic in Haidian District were the focus: "urban villages–blue–green space" oriented by residents' health, "urban villages–real estate" oriented by domestic finance, and "urban villages–municipal facilities" oriented by public services. They were further analyzed using the example of Chuanying Village, located in the middle of Shiziqing Town, as shown in Figure 9.

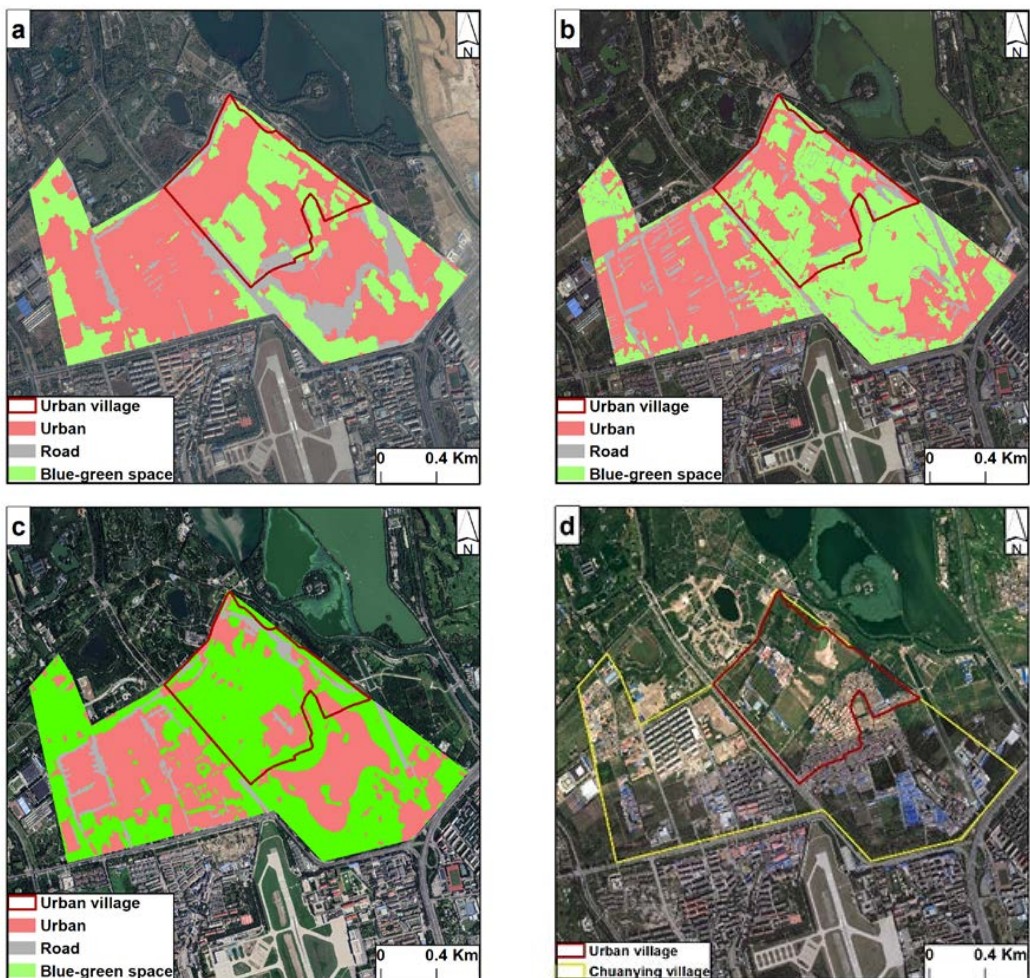

**Figure 9.** Land use recognition result of the urban village in Chuanying Village: (**a**) 2013, (**b**) 2015, (**c**) 2020, and (**d**) remote sensing image on 27 September 2010.

The name of Chuanying Village means the village where ships were moored. Chuanying Village stands at the upstream hub of the entire Beijing water system which connects to the inner lakes of the Summer Palace and Yuanmingyuan, as well as major rivers of

Beijing such as the Qing River and the Nanchang River. This is why it was one of the most significant garrisons for the Qing Dynasty's naval forces in the 18th century, and families of the naval forces farmed and lived here until modern times. Therefore, Funaying Village has a long historical heritage and cultural value.

The entire area of Chuanying Village is adjacent to the cultural and scenic attraction, the Summer Palace. It is bordered by residential areas to the south and west. In 2010, it was still an urban village with an area of 3.41 km$^2$ (Figure 9d). It was partially demolished in 2013. Moreover, it was transformed into Beiwu Property, Zhongwu Property, Zhongwu Park, Chuanying Park, Yuanwaiyuan Park, and the Beijing South–North Water Diversion Project Construction Management Center in 2020. The entire area of Chuanying Village is a paragon of urban renewal land use change in Haidian District.

### 5.1. Resident Health Orientation: Transformation of Urban Villages into Blue–Green Spaces

The "three hills and the five gardens" area is a crucial zone in the conservation system of Beijing's historic and cultural cities, including the world-famous Summer Palace and Yuanmingyuan. This area plays an overarching role in the ecological conservation and cultural heritage of Beijing. Chuanying Village stands in the middle of the "three hills and the five gardens" area. Therefore, the fine-scale regeneration from urban villages to blue–green spaces in Chuanying Village plays a complementary role in the efficient use of space, public services, and recreation. In the whole urban system, the wellbeing of all ages, particularly the elderly and children, is enhanced by installing trails and cycling paths. Chuanying Village transformed the urban village into three parks, including Zhongwu Park, Chuanying Park, and Yuanwaiyuan Park. On the one hand, the village is connected to the cultural site of the Summer Palace to create a cultured zone with both ecological and tourism values. On the other hand, it follows the historical and cultural heritage of the district as a traditional shipping base to create a leisure and recreational space with historical values.

### 5.2. Local Financial Orientation: Transformation of Urban Villages into Real Estate

Land finance is an important source of revenue for local governments. The demolition and relocation of urban villages require a large amount of demolition and relocation costs. Thus, relying on revenue from real estate or even in situ construction of resettlement housing is also an unavoidable path to achieve the upgrading of urban villages under cost constraints. Chuanying Village has two real estate development models: Zhongwu Property, at the north of the site, is sold as a high-end commercial villa neighboring the Summer Palace, whereas Beiwu Property in the west is a resettlement house for farmers relocated in situ for demolition and upgrading. After reasonable construction planning, Beiwu Property occupies only approximately 10% of the original site but accommodates most of the interested relocated population.

### 5.3. Public Service Orientation: Transformation of Urban Villages into Municipal Facilities

Parts of the urban villages are also transformed into facilities, such as urban roads and bridges, to serve the public infrastructure of the city effectively. In the eastern part of Chuanying Village, 18% of the Chuanying Village blocks were transformed into the Construction Management Center of the South-to-North Water Diversion Project in Beijing in 2020 by relying on the original Jing-Mi diversion canal. Thus, this part of Chuanying Village has become one of the most important hubs ensuring year-round water supply in Beijing. This transformation of urban villages into municipal facilities under the Orientation of public service not only enhances the water supply function and capacity of Haidian District, but also ensures the entire water security of Beijing's 21.89 million residents in the capital city of China.

## 6. Conclusions

In this study, the spatial and temporal patterns of land use changes in urban villages were quantitatively presented to answer the following question: What has become of the disappearing urban villages? The urban renewal logic of land use transformation in urban villages is reflected in a hierarchical machine learning identification method using three phases of remote sensing imagery from GF-1 and performing the scene-based RF classification. The results show that in 2013, the overall scale of Haidian urban village blocks was 10.46 km$^2$, showing a spatial pattern along traffic arteries; in 2015, it declined to 10.11 km$^2$. Moreover, the size of the urban village blocks in 2020 decreased to 1.02 km$^2$, 9.75% of that in 2013. Only a few urban village blocks were distributed in clusters in Qinghe Subdistrict, Qinglongqiao Subdistrict, and the Four Seasons Green Subdistrict. With Chuanying Village as an example, three levels of urban renewal logic were revealed: "urban village–blue–green space", "urban village–real estate", and "urban village–municipal facilities".

The findings above also provide creative inspiration for future related research, as discussed below:

1.  In terms of identification methods, a scene-based recognition method was used to identify urban villages in Haidian District based on the remote sensing data of GF-1 satellite. This method uses the high-level semantic information of urban villages to identify urban village areas by extracting, counting, and analyzing the overall features of buildings within the blocks. However, using the scene-based identification method for urban village identification has certain flaws. First, the suitability of the scene, namely, the correct division of the blocks, is extremely important for the method. Second, certain tiny urban villages are not correctly identified during the recognition process, mainly because of the weak influence of small urban village buildings on the overall features of the whole block. Thus, the overall features of the block still tend to be urban buildings. Therefore, the fine delineation of scenes must be focused on in future research. Given the limited capability of optical images in urban mapping, multidimensional identification of urban villages can also be considered in the future by combining multisource data, such as POI data and population data, including the height of buildings and the development of multifunctional buildings. In terms of scale, this paper focuses more on the trend changes and spatial and temporal patterns of urban villages as a whole. Changes within the urban village, including changes in building density and floor area ratio, will likewise affect the attributes and functions of the urban village as a migrant cluster. Such a phenomenon may not be ruled out: part of the peripheral land of an urban village is given away (perhaps farmland), but the building density and floor area ratio in its center increases. Although the area of the urban village block is reduced in this case, it has a stronger capacity to accommodate migrants and therefore a broader employment hinterland that can be radiated externally. Therefore, higher resolution remote sensing images, more accurate remote sensing interpretation techniques, block-based demographic data and related economic and social data should be used to resolve the accommodation capacity and its radiating capacity of internal urban villages as migrant clusters.
2.  In terms of the transferability of the research, the scene-based approach has great potential for complex urban structure detection, particularly in modeling and portraying complex scenarios, which also play an active role in transferability studies. Due to Beijing's capital function and Beijing's high level of urbanization, Haidian District exhibits a relatively ideal state of urban village regeneration—only a decrease in urban villages without the emergence of new urban villages either in terms of area or location. The emergence of new urban villages will obviously present a more complex land use flow and a more integrated urban village renewal pattern in the study of the overall urban system. As urbanization in China is still growing at a high rate, new periurban villages may gradually become urban villages as the urban fringe sprawls. Therefore, the question answered in this study, i.e., "What has become of the disappearing urban villages?" can be extended to "How do new urban villages

emerge and how do they transform?" in further research. The type of research can be expanded from mature megacities to medium and large cities undergoing rapid urbanization. The scope of research will be expanded from Beijing to other research regions worldwide to provide experiences and insights on informal settlements and the development of urban sprawl in Africa and Latin America.

3. In terms of policy revelation, given the advent of urban renewal in urban planning practice, a single flow of land to real estate is no longer feasible in terms of local finance and urban ecology. An integrated, multieffective, and systematic approach to renewal that combines financial gains and public services is being sought instead. Urban green parks have become an important flow of land in urban villages in urban regeneration because of the positive role of blue–green space in improving the health of residents, the quality of life of citizens, and the urban environment and climate regulation of cities. These parks also contribute to the green and low-carbon sustainable development of cities. On the other hand, given the detailed analysis of urban village land flows in this study, local governments are already realizing different and differentiated urban functions through multiple urban village urban renewal land flows in practice. We are committed to introducing a product portfolio for urban renewal in urban villages that balances the fiscal budget and residents' wellbeing in future research. In particular, we are devoted to establishing an urban renewal model for urban villages with the view of maintaining a balance between fiscal budget and residents' wellbeing by measuring the revenue and cost of real estate, the value and cost of landscape and ecology brought by the construction of green parks, and the value and cost of public services brought by large-scale municipal projects. Moreover, the externalities, such as the impact of land prices in the surrounding areas before and after the transformation and the revenue brought by the accessibility of transportation roads, are considered. The model is used to simulate different shares and proportions of land flow in urban villages under various scenarios of location and economic development. It also provides guidance and reference for global urban renewal policymakers and urban planners.

**Author Contributions:** Conceptualization, H.W. and W.Q.; methodology, H.W. and W.Q.; software, H.W. and Y.C.; validation, Y.C. and H.W.; formal analysis, H.W. and Y.C.; investigation, W.Q.; resources, W.Q.; data curation, H.W. and Y.C.; writing—original draft preparation, H.W. and Y.C.; writing—review and editing, H.W. and W.Q.; visualization, H.W. and Y.C.; supervision, W.Q.; project administration, W.Q.; funding acquisition, W.Q. All authors have read and agreed to the published version of the manuscript.

**Funding:** This research and the APC was funded by the National Natural Science Foundation of China (Grant No. 42171237).

**Data Availability Statement:** The data presented in this study are available on request from the author.

**Conflicts of Interest:** The authors declare no conflict of interest.

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
