# Peer review of "The Vanishing and Renewal Landscape of Urban Villages Using High-Resolution Remote Sensing: The Case of Haidian District in Beijing"

_remotesensing, doi:10.3390/rs15071835_

Round 1

Reviewer 1 Report

Good for the publication in the present form

Author Response

Point 1: Good for the publication in the present form,

Response 1: Thank you for your acknowledgement and support of this study. Further improvements of this paper were made based on the suggestions of two other reviewers.

Reviewer 2 Report

The research is very interesting and can be applied to other typologies of urban tissue, not only urban villages.

The results can be of interest to non-specialists as well. 

Just a few minor remarks:

Line 32 - short roads or narrow roads?

Line 33-35 - please explain better Because rapid urban expansion and low prices for land do not automatically trigger epidemic diseases ... etc. 

Line 53 and line 55 - better use "the migrants" and not "these migrants". I sounds condescendent. 

Author Response

Point 1: Line 32 - short roads or narrow roads?

Response 1:

Thank you for your approval and support of this article. Thank you for your rigorous remark, which greatly improves the academic rigor of this paper. One of the typical characteristics of urban villages is that the width of roads does not meet the requirements of urban road construction and is very narrow. This leads to inconvenient traffic as well as fire hazards within urban villages.

Therefore, we have changed "short roads" to "narrow roads" in Line 32 according to your suggestion.

Point 2: Line 33-35 - please explain better Because rapid urban expansion and low prices for land do not automatically trigger epidemic diseases ... etc.

Response 2:

Thank you for your pertinent advice, which has greatly improved the logic of this article. The original Line 33-35 do not explain very well how rapid urban expansion and low land prices have led to the urban village epidemic. In fact, it is due to the architectural features of urban villages which are mentioned in the previous sentence. Rapid urban expansion and low-priced land lead to cramped buildings and disorderly streets within urban villages, hence numerous entrances and exits and substandard living and production facilities. This has led to firefighting problems and the spread of epidemics, respectively.

Based on this, we make the following changes to Line 33-35:

Replace " Given the rapid jump in urban expansion and the cost of urban land acquisition in urban development, urban villages contribute to the spread of epidemic diseases, fire, and other safety hazards " with " There are negative effects of this physical characteristics of urban villages arising from the rapid jump in urban expansion and the cost of land acquisition in urban development. First, there are many streets and alleys that are intricate and complex, so entrance and exit routes are numerous and irregular, leading to fire and other safety problems; in addition, living and production facilities are relatively rudimentary, and domestic waste and sewage are rarely properly disposed of, leading to the spread of sanitation and epidemics in public places."

We would like to thank you again for taking the time to review our manuscript. Thank you very much for your attention and time.

Reviewer 3 Report

The paper is focused on the recognition of land use change of urban villages starting from high-resolution remote sensing images of 2011, 2015, and 2020, assessing changes before and after urban renewal by using a hierarchical machine learning recognition method based on scene-based and random forest (RF) classification.

The spatial identification of urban villages has become a problem considered in remote sensing and GIS disciplines.

The topic is relevant since it is related to urban assessment and possible regeneration actions by applying remote sensing analysis, a very topical tool nowadays.

·        It is suggested to add a keyword able to identify the applied assessment procedure.

The aim is to use high-resolution remote sensing images of the three time phases (2013, 2015, and 2020) to reflect the land use changes of urban villages before and after urban renewal as a basis for urban regeneration plans.

In the introduction, the research framework is clearly outlined (informal settlements: architectural, urban and social impacts, and urban renewal issues).

The paper analyses previous studies related to density-based methods, remote and social sensing data and partition-strategy-based framework, aiming at bridging the gap on the dynamic spatial and temporal patterns of urban villages (not studied for a certain time series span). In this direction, previous land-use and land-cover change’ assessments procedures are analysed, discussing advantages in using machine learning.

·        However, the State of the Art in applying machine learning methods to related fields (landscape, urban areas) is addressed superficially. It would be of interest for the research to assess some previous results (urban scale / kind of data / use of these data / possible impact for the current research assessment).

The study area and available datasets are shortly described in paragraph 2.1; additional data would be useful (buildings typologies, critical areas, density, footprints, etc.) and additional pictures to highlight – in a top view with possible legend to identify different urban areas – patterns and features.

The Methodology includes features extraction (texture, spectral, and geometric information), areas are classified into three types of land use: urban buildings, roads, and blue-green space.

·        Some graphical outputs from scene segmentation should be added.

·        In general, pictures and graphical schemas should be added in order to better support the discussion, according to the performed steps.

Out of 32 references, 11 are dated from the last five years. Cited references are relevant to the research.

The paper is divided into sections that make the research approach clear and consistent.

The methods are adequately described and the overall experimental process is generally explained.

Typos

- line 86: a word seems missing;

- check line 221.

Author Response

Point 1: It is suggested to add a keyword able to identify the applied assessment procedure.

Response 1:

Thank you for your detailed suggestions for this article. Adding a method-related keyword would truly highlight the urban village identification technique in this paper, so we will add “scene-based” as a new keyword.

Point 2: However, the State of the Art in applying machine learning methods to related fields (landscape, urban areas) is addressed superficially. It would be of interest for the research to assess some previous results (urban scale / kind of data / use of these data / possible impact for the current research assessment).

Response 2:

Your detailed suggestions on the research methodology review part of this paper are greatly appreciated. To enrich the methodological review part, we have added a paragraph to show the evaluation of the effectiveness of the existing research on urban village identification and the current status of the application of machine learning methods in the field of urban village identification.

The added paragraph is as follows (Section 1. Introduction., Line 119):

The identification of urban villages is an enormous challenge in complex urban environments [25]. The traditional pixel-based extraction methods of remote sensing in-formation are not applicable to urban village extraction, since individual urban village pixels do not differ much from other urban buildings [26]. Therefore, it is necessary to exploit the semantic information surrounding urban villages in order to extract it accurately. Extensive research on high-resolution remote sensing information extraction tends to obtain semantic objects with similar spectrum or texture information by semantic segmentation of images, which leads to the classification of these objects. Although this object-based approach is effective for distinguishing vegetation, buildings, and other landscape forms, urban village and urban building objects with similar characteristics cannot be further distinguished [19]. Here, the scene-based method shows strong superiority in urban village identification. The base classification unit of the scene-based approach is not a single object, but a scene that aggregates information of multiple semantic objects [27]. Because of the particular historical background, the urban village scenes are vastly different from the traditional urban building scenes in terms of general pat-terns and building density, which also shows certain regional variations [28]. Consequently, it is effective to identify urban villages through the following hierarchical steps: First, urban building objects are extracted through an object-based approach. Then, the semantic information (spectrum, texture and geometry, etc.) of all building objects in the urban area scene is statistically calculated. Finally, machine learning methods are applied for identification.

25.Niu, N.; Jin, H. Integrating multiple data to identify building functions in China’s urban villages. Environ. Plan B Urban Anal. City Sci. 2021, 48, 1527–1542. [CrossRef]

26.Zhu, Y.; Geiss, C.; So, E.; Jin, Y. Multitemporal Relearning With Convolutional LSTM Models for Land Use Classification. IEEE J. Sel. Top Appl. Earth Obs. Remote Sens. 2021, 14, 3251-3265. [CrossRef]

27.Liu, H.; Wu, C. Developing a Scene-Based Triangulated Irregular Network (TIN) Technique for Individual Tree Crown Reconstruction with LiDAR Data. Forests 2020, 11, 28. [CrossRef]

28.Gao, Y.; Shahab, S.; Ahmadpoor, N. Morphology of Urban Villages in China: A Case Study of Dayuan Village in Guangzhou. Urban Sci. 2020, 4, 23. [CrossRef]

Point 3: The study area and available datasets are shortly described in paragraph 2.1; additional data would be useful (buildings typologies, critical areas, density, footprints, etc.) and additional pictures to highlight – in a top view with possible legend to identify different urban areas – patterns and features.

Response 3:

Thank you very much for your pertinent suggestions. The visualization of the urban village features and patterns will greatly increase the comprehensibility of this paper. Therefore, we have created a top view of the local features of the urban village and a comparison with the non-urban village neighborhoods. The comparison image is added after the detailed description statements about the characteristics of urban village blocks and the differences with non-urban village blocks(Line 172).

Figure 2. Sample examples of urban village blocks and non-urban village blocks.

Point 4: Some graphical outputs from scene segmentation should be added. In general, pictures and graphical schemas should be added in order to better support the discussion, according to the performed steps.

Response 4:

Your advice is certainly spot on. As a paper in the field of remote sensing, more graphical output about the identification method is indispensable in order to support the research procedure better. We have added a set of figures for showing the semantic segmentation process of high-resolution remote sensing images (Figure 4.) in Section 3.2. Building Information Extraction.

Fig.5 Building Information Extraction (a) local high-resolution images, (b) image objects, (c) results of building information extraction

Point 5: Typos: 1. line 86: a word seems missing; 2.check line 221.

Response5:

Thank you for your meticulous advice. Line 86 of the sentence “Spatial governance is a tool for social governance. In order to further the spatial governance of urban villages, spatial identification of urban villages has become a problem considered in remote sensing and GIS disciplines” was checked by the software Grammarly and my native English speaking colleagues and no grammatical problems were found. It may be that the verb version of the word "further" is less used today.

As for the problem in line 221, after carefully checking the grammatical issues, we believe that the lack of clear labeling of the math letters caused the problem. Therefore, we changed the math operation letters and symbols in lines 221-227 to Italian italics, as follows:

  • Randomly sample the training set m times to obtain a sample set D with sample size m.
  • Randomly select k attribute features among all attribute features and select the best segmented attribute features as nodes to construct the decision tree.
  • Repeat the above two steps T times, i.e., T decision trees are built.
  • These T decision trees form an RF and vote on which category the data finally belongs to by voting.

We would like to thank you again for taking the time to review our manuscript. Thank you very much for your attention and time.
